# DIFFUSER: DIFFUSION VIA EDIT-BASED RECONSTRUCTION

**Machel Reid**
Google Research*
machelreid@google.com

**Vincent J. Hellendoorn**
Software and Societal Systems Department
Carnegie Mellon University
vhellendoorn@cmu.edu

**Graham Neubig**
Language Technologies Institute,
Carnegie Mellon University
Inspired Cognition
gneubig@cs.cmu.edu

## ABSTRACT

In text generation, models that generate text from scratch one token at a time are currently the dominant paradigm. Despite being performant, these models lack the ability to *revise existing text*, which limits their usability in many practical scenarios. We look to address this, with DIFFUSER (**Diffus**ion via **E**dit-based **R**econstruction), a new edit-based generative model for text based on denoising diffusion models – a class of models that use a Markov chain of denoising steps to incrementally generate data. DIFFUSER is not only a strong generative model in general, rivalling autoregressive models on several tasks spanning machine translation, summarization, and style transfer; it can also perform other varieties of generation that standard autoregressive models are not well-suited for. For instance, we demonstrate that DIFFUSER makes it possible for a user to condition generation on a prototype, or an incomplete sequence, and continue revising based on previous edit steps.

## 1 INTRODUCTION

Revision and editing are central to how humans produce content; we write and revise emails and papers, gradually produce works of art, and iterate on plans for a project. Despite this, the most dominant paradigm in text generation is purely autoregressive, producing text left-to-right in a single pass (Bengio et al., 2003). Although models employing this single-pass form of generation are highly performant, they are limited by the inability to refine existing text. To address this, we propose DIFFUSER: **Diffus**ion via **E**dit-based **R**econstruction, a flexible method to apply edit-based generative processes to arbitrary text generation tasks. Specifically, we take inspiration from diffusion models (Sohl-Dickstein et al., 2015; Ho et al., 2020), generative models that generate by way of incremental denoising steps, and adapt this approach to the text generation paradigm with a formulation similar to natural editing processes.

Prior work on text generation either focuses on improving the performance of standard autoregressive (AR) models through larger models and datasets (Vaswani et al., 2017; Sutskever et al., 2014; Radford et al.; Brown et al., 2020) or on proposing new, non-autoregressive approaches (Gu et al., 2017; Ghazvininejad et al., 2019; Gu et al., 2019) to improve general modes of text generation. A thus far separate line of models has taken the perspective of modeling text edits for specific tasks: e.g. style transfer (Reid & Zhong, 2021; Malmi et al., 2020), sentence fusion (Malmi et al., 2019), and grammatical error correction (Dale & Kilgarriff, 2011). DIFFUSER unifies these two perspectives by enabling edit processes to be applied to general purpose text generation without compromising performance or requiring external supervised data (Guu et al., 2018). This design enables it

---

*Work done partially while at the University of Tokyo

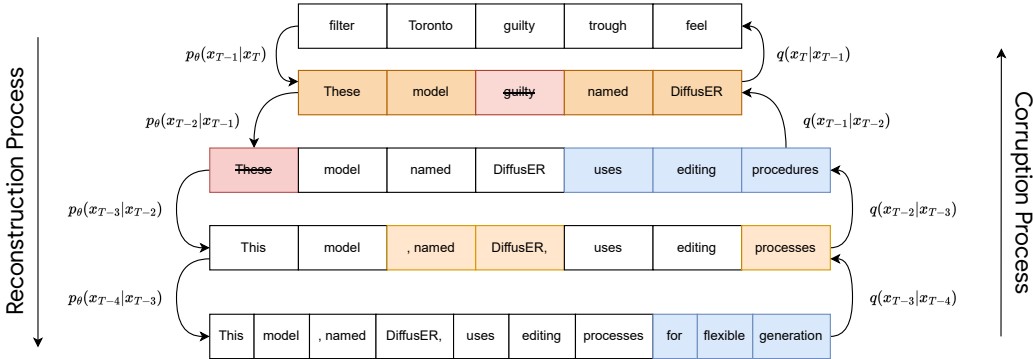

Figure 1: DIFFUSER's text generation process. Orange represents replacements, blue represents insertions, red represents deletions, and white represents keep operations. This process largely imitates a natural editing process (Reid & Neubig, 2022).

to both generate and edit text, including externally produced content, a natural extension of the text generation paradigm.

DIFFUSER models text generation as a series of diffusion steps at the token level. This form of generation allows us to develop a synthetic formulation of natural editing processes (Reid & Neubig, 2022) using edit-based corruption and reconstruction. Our method starts from an arbitrary sequence (either a prototype generation, randomly sampled tokens, or a null sequence) and progressively edits it into the final sequence guided by the Levenshtein edit operations of INSERT, DELETE, KEEP, and REPLACE as shown in Figure 1. This enables flexible editing in a range of contexts, including machine translation, summarization, style transfer, while also allowing for the possibility of taking outside input to guide and constrain generation.

Learning these edit-based diffusion processes required several innovations over standard autoregressive and MLM-style iterative generation approaches (Ghazvininejad et al., 2019; Austin et al., 2021; Savinov et al., 2022), including forming edit-based corruption and reconstruction processes for training (Sec 3), as well as techniques to improve the quality of decoding sequences across both timesteps and token-level generations (including 2D beam search; Sec 3.6, Sec 3.5).

To demonstrate the effectiveness of DIFFUSER, we test our method on three text generation tasks: machine translation, abstractive summarization, and text style transfer, and show on-par or improved performance compared to purely autoregressive, single-pass and non-autoregressive methods. We also provide qualitative samples of the edit processes learned by the models in different settings and analyses on training and inference speeds, as well as the relationship between edit steps and performance.

Overall, we demonstrate the potential of edit-based generative models to offer 1) more performant generation, 2) greater interactivity between different models (as we can now perform edits in the discrete space on model generated output), and 3) more flexible/controllable generation.

## 2 BACKGROUND

DIFFUSER operates at the intersection of text generation, editing processes, and diffusion models. We first provide the background and intuition of these three techniques.

### 2.1 TEXT GENERATION

Most text generation models used in NLP today are autoregressive in nature. In this paradigm, given a sequence $\mathbf{s} = [\mathbf{s}_0, \mathbf{s}_1, \ldots, \mathbf{s}_N]$, one can model the likelihood of the entire sequence $P(\mathbf{s})$ by modeling the probability of predicting each token in an autoregressive, often left-to-right, manner. This formulation, where the likelihood of a token $p(\mathbf{s}_t)$ is conditioned on its predecessors $\mathbf{s}_{<t}$, is

shown below (Bengio et al., 2003):

$$P(\mathbf{s}) = \prod_{i=0}^{N} p(\mathbf{s}_t | \mathbf{s}_{t-1}, \mathbf{s}_{t-2}, \dots, \mathbf{s}_0) \tag{1}$$

Models trained with this objective can then be sampled from, or searched over (e.g. using beam search), to provide generations in downstream tasks such as machine translation or summarization.

Non-autoregressive models (Gu et al., 2017) are a different variety of generative models, in which a sequence is generated in a single pass (removing the autoregressive conditioning on previously generated tokens) with multiple revision-level passes, often in the name of efficiency.

## 2.2 EDITING PROCESSES

Editing processes (Reid & Neubig, 2022) are a paradigm for modeling text by way of incremental revisions, taking inspiration from the the way humans generate text. Specifically, let $X = \{\mathbf{x}_0, \mathbf{x}_1, \dots, \mathbf{x}_R\}$ be a series of $R$ versions of a document, where $\mathbf{x}_0, \mathbf{x}_i, \mathbf{x}_R$ represents the initial, intermediate (at timestep $t$), and final/current state of a document, respectively. Using editing processes, we can model the probability of this series of documents versions occurring consecutively as follows:

$$p(X) = \prod_{i=0}^{R} p(\mathbf{x}_i | \mathbf{x}_0^{i-1}) \tag{2}$$

With this formulation, editing processes can also be used to calculate the probability of only the final document while taking into account previous revisions, which is not possible in the traditional text generation setup as intermediate revisions are not explicitly known, using the equation below (Reid & Neubig, 2022).

$$p(\mathbf{x}_R) = \sum_{\tilde{X} \in \{\tilde{\mathbf{x}}_0^R | \tilde{\mathbf{x}}_R = \mathbf{x}_R\}} p(\tilde{X}). \tag{3}$$

## 2.3 DIFFUSION MODELS

We now make the connection between editing processes and diffusion models (Sohl-Dickstein et al., 2015; Ho et al., 2020). Continuous diffusion processes are commonly applied in computer vision tasks to iteratively convert a sample of noise into an image. This can be seen as an edit process in which the model iteratively *edits* a noisy image to bring it closer to a final, complete image. These continuous diffusion models are often trained by modeling a Markov chain $\mathbf{x}_T \dots \mathbf{x}_t \dots \mathbf{x}_0$, where $\mathbf{x}_0$ represents the original image and $\mathbf{x}_T$ represents Gaussian noise. This chain is typically produced by incrementally adding Gaussian noise to $\mathbf{x}_t$ to form $\mathbf{x}_{t+1}$ (known as the *forward* or *corruption* process), wherein a model parameterised by $p_\theta$ is trained to *reverse* (or "*denoise*") this process to form the chain $\sum_{i=1}^{T} p_\theta(\mathbf{x}_{t-1} | \mathbf{x}_t)$.

Analogized to text, this allows us to formulate natural edit processes as a discrete diffusion process in which a null string or a prototype is iteratively edited into free form text. Our DIFFUSER method (Figure 1) takes inspiration from this process, but parameterises the corruption process by way of sampled discrete edit operations applied over a discrete sequence of tokens. The success of our method supports the findings in the vision domain Bansal et al. (2022), where it is found that diffusion models can learn to invert arbitrary transformations.

Previous work in diffusion models has largely focused on computer vision (Ho et al., 2020; Austin et al., 2021), in which the diffusion process is applied to raw image values. Within the context of natural language, both discrete diffusion models using only replacement operations (either applied to random tokens or masked tokens) (Savinov et al., 2022; Austin et al., 2021), and continuous diffusion over word embeddings (Li et al., 2022) have been proposed. Our model is a more flexible approach, using all four edit operations, towards diffusion models when compared with this work owing to its edit process formulation, and is also more compatible with current models (e.g. AR bootstrapping).

## 3 DIFFUSER

DIFFUSER, being a diffusion-based method, has two main procedures: corruption and denoising. Unlike previous work (Ghazvininejad et al., 2019; Savinov et al., 2022; Gu et al., 2019) in which this procedure is relatively inflexible (e.g., due to length restrictions and/or using continuous representations for the basis of the diffusion process), both our corruption process and denoising process are based on Levenshtein operations, allowing our model to learn to take advantage of the flexibility of text editing when generating.

### 3.1 EDIT OPERATIONS

Given the central role of the Levenshtein edit operations in our models, we provide a brief overview of each operation and its role in the editing process. We use Figure 1 as a guide when explaining each operation.

**INSERT:** The insertion operation is used to add new text to a sequence. For example in Figure 1, "*uses editing processes*" is added by *DiffusER* at timestep $x_{T-2}$.

**DELETE:** The deletion operation erases existing text. In Figure 1, this is shown when "*These*" gets deleted at timestep $x_{T-2} \rightarrow x_{T-3}$.

**REPLACE:** The replacement operation works overwriting existing text with new text. This is shown in Figure 1 at step $x_T \rightarrow x_{T-1}$ where "*filter Toronto guilty trough feel*" is replaced by "*These model guilty named DiffusER*".

**KEEP:** The keep operation ensures that a portion of the text remains unchanged into the next iteration. This is illustrated in timestep $x_{T-2} \rightarrow x_{T-3}$ where "*model named DiffusER*" is kept.

### 3.2 EDIT-BASED CORRUPTION

The four Levenshtein edit operations described above allow us to transform any arbitrary sequence of tokens into another. This is in contrast to iterative mask replacement, which can only introduce new tokens (Ghazvininejad et al., 2019; Austin et al., 2021; Savinov et al., 2022). For every timestep $i$, corruption process $q(\mathbf{x}_i | \mathbf{x}_{i-1}; \mathcal{E}_t, \mathcal{E}_l)$ is parameterized by two distributions: the distribution over edit types $\mathcal{E}_t$ (e.g. 60% keep, 20% replace, 10% delete, 10% insert), and the distribution over edit length $\mathcal{E}_l$. The latter can be parameterized by any distribution over non-negative integers, such as a uniform distribution or a Poisson distribution. For instance, to learn a deletion operation in the reconstruction process, we insert randomly sampled distractor tokens, whereas, to learn an insertion operation we delete a subset of tokens contained in the sequence.

### 3.3 EDIT-BASED RECONSTRUCTION

Our generative process is trained via the ***Edit-based Reconstruction*** (ER) process. ER can be thought of as the opposite of our corruption process, in which we need to find the appropriate edit operations to transform $\mathbf{x}_T$ to $\mathbf{x}_0$, by way of $\mathbf{x}_{T-1}, \ldots, \mathbf{x}_1$.

That is, given a corrupted sequence $\mathbf{x}_T$, we aim to learn the process by which we can reverse the corruption in the following form.

$$P_\theta(\mathbf{x}_0) = \prod_{t=0}^{T} p_\theta(\mathbf{x}_{t-1} | \mathbf{x}_t) \tag{4}$$

Given that, we model the likelihood of each timestep $\mathbf{x}_t$, this can also be referred to as an edit process (Reid & Neubig, 2022). As we include an edit process in our model and use Levenshtein tags for editing, one can think of ER as two distinct steps: identify which edits should take place (tagging process) and deciding which tokens should go in these positions (generative process). This decomposition is shown here:

$$p_\theta(\mathbf{x}_{t-1} | \mathbf{x}_t) = p_\theta^{\text{tag}}(\mathbf{e}_t | \mathbf{x}_t) p_\theta^{\text{gen}}(\mathbf{x}_{t-1} | \mathbf{x}_t, \mathbf{e}_t) \tag{5}$$

where $p_\theta^{\mathrm{tag}}$ parameterises the tagging model to estimate the likelihood of producing a given set of Levenshtein edit operations $\{\textsc{Insert}, \textsc{Delete}, \textsc{Keep}, \textsc{Replace}\}$ given $\mathbf{x}_t$, and $p_\theta^{\mathrm{gen}}$ parametersies the generator model given sequence $\mathbf{x}_t$ and edit operations $\mathbf{e}_t$. This decomposition via edit-operations allows the generation process to be more controllable and more flexible as it allows up to explicitly specify edit types associated with tokens to be edited, rather than leaving both processes to be implicit.

## 3.4 Implementing DiffusER with Transformers

When implemented with Transformers (Vaswani et al., 2017), DiffusER consists of two components: a tagger and generator. The tagger, a transformer network, is trained using cross-entropy loss over the ground-truth tag types to predict the edit operations that should be applied to the sequence, in preparation for the next generation step. Then, in the generation step, after removing tokens selected for deletion, we sum a learned embedding to insert and replace types and generate the inserted and replaced sequences autoregressively. Following this, we feed the output of this diffusion step into the tagger and perform another diffusion step. One step of this process can be compared to the reconstruction process used in Aghajanyan et al. (2022).

## 3.5 Decoding Methods

DiffusER has an inherently different generation process from a standard autoregressive language generation model—in addition to operating on a sequence/token level (in which generation is composed of generating individual tokens in a single-revision; *intra-revision*), we also operate on a *revision* level (in which the text is expanded across diffusion steps, *inter-revision*). This allows us to experiment with different methods for decoding on both the intra-revision (single sequence level) and inter-revision levels (multiple version level), which we explain below.

**Beam Search**    One method for decoding is to perform beam search over $b$ hypotheses at every step on the output of our autoregressive generator (intra-revision level), while performing greedy decoding at the inter-revision level. Although being conceptually straightforward, this method has the limitation of not searching over the inter-revision space (despite revisions being a key component of our approach).

**2D Beam Search**    We propose 2D beam search, in which we extend beam search as it is applied to token-level autoregressive generative models, and perform beam search using both an intra-revision width of $b$ and an inter-revision beam width of $r$. This allows us to perform search on the inter-revision level, which we find results in better downstream performance, but increases the beam count to $r \times b$ beams. Assuming a fixed sequence length and maximum number of diffusion steps, we would decode as follows: We first use beam search with width $b$ at the token level and take the $r$ most likely candidates (measured with log-likelihood). These $r$ candidates are then fed to the next step of the diffusion model, wherein for each of $r$ hypotheses the next diffusion step is performed with the token-level generator decoding with beam width of $b$. This leads us to have $r \times b$ candidate hypotheses, of which we take the top $r$. This process repeats for each diffusion step thereafter.

**Nucleus Sampling**    To improve the diversity of generations, we also consider a nucleus sampling based approach, where at every timestep $\mathbf{x}_t$, we use nucleus sampling (Holtzman et al., 2019) with $p = 0.6$ to sample each token autoregressively at the intra-revision level, and greedily decode at the inter-revision level (i.e. no search or sampling is performed over multiple diffusion steps).

## 3.6 Decoder Initialization Techniques

Since our model is based on edit processes, it offers flexibility in terms of the discrete sequence from which to initialize the text generation. Previous work on non-autoregressive translation often starts with `[MASK]` tokens (Ghazvininejad et al., 2019), a null string (Gu et al., 2019) or random tokens (Savinov et al., 2022). We include the latter two methods in our experiments, in addition to (1) experimenting with an AR Bootstrap, in which we learn to bootstrap from text generated by a purely autoregressive model, and (2) proposing to use the source-side text as an initial state for the DiffusER decoder.

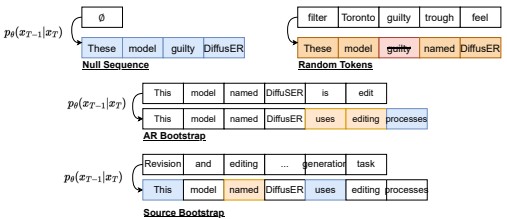

Figure 2: Figure illustrating bootstrapping methods for decoding.

**Null Sequence** In this setting, we simply initialize DIFFUSER with a null string, in which the first edit is constrained to be insertion.

**Random Tokens** In this setting, we initialize DIFFUSER with a series of random tokens, following (Savinov et al., 2022). The model then learns to edit this random sequence.

**AR Bootstrap** We bootstrap the reverse diffusion process by taking the output of DIFFUSER constrained to generate autoregressively (essentially mimicking a standard autoregressive generator). We then use DIFFUSER to further edit the output of this operation.

**Source Bootstrap** In a sequence-to-sequence setting, we can also generate by bootstrapping using the source text, by setting $x_T$ to be equivalent to s. As we show in later sections, this is particularly useful in tasks such as summarization in which the output can be easily formulated as an editing version of the input.

## 4 EXPERIMENTS

### 4.1 MODELS

**DIFFUSER** We instantiate DIFFUSER with two separate Transformer models for the tagger and generator. We use the Transformer-base encoder-decoder (Vaswani et al., 2017) architecture, with 6 layers, for the a hidden dimension of 512, feedforward dimension of 2048, 8 attention heads, and dropout $p = 0.3$.

**Baselines (MT & Summ)** We use several Transformer baselines from previous literature for our various tasks. We include a conventional 6-layer encoder-decoder Transformer model from Vaswani et al. (2017), as well as models proposed in related work from the non-autoregressive generation literature: Levensthein Transformer (Gu et al., 2019), CMLM (Ghazvininejad et al., 2019), DisCo (Kasai et al., 2020a), Imputer (Saharia et al., 2020), and SUNDAE (Savinov et al., 2022).

### 4.2 TASKS

**Machine Translation** We use the WMT'14 English-German dataset for our machine translation experiments. We use the same preprocessing and post-processing steps as Ghazvininejad et al. (2019). Unlike the standard in non-autoregressive translation work (Zhou et al., 2019), we focus on using the gold machine translation data instead of distilled data. We use a Poisson distribution $\mathcal{E}_l(\lambda = 3)$ over edit operation lengths in our corruption process. Note that we compute the edit operations over words rather than tokens. For this task, as well as the following ones, we use 12 diffusion steps, $b = 5$, and $r = 3$ for beam search, and $\mathcal{E}_t(60\% \text{ KEEP}, 20\% \text{ REPLACE}, 10\% \text{ INSERT}, 10\% \text{ DELETE})$ based on numbers from preliminary experiments.

**Summarization** We also benchmark on the CNN/DailyMail dataset for summarization (Nallapati et al., 2016). Summarization is different in nature from machine translation in that it can be described as more conducive to edits as a good summary tends to preserve many parts of the input. We use the same post-processing steps as See et al. (2017). We use a Poisson distribution $\mathcal{E}_l(\lambda = 8)$ over edit operation lengths in our corruption process (to roughly model sentence boundaries).

**Text Style Transfer** We perform experiments using the Yelp (Shen et al., 2017) dataset for the unsupervised text-style transfer task. We compare against methods such as Tag-and-Generate (Madaan et al., 2020), Masker (Malmi et al., 2020), and LEWIS (Reid & Zhong, 2021). In contrast with machine translation and summarization, text style transfer datasets are often unaligned (i.e. without

| Model | En-De (MT) | CNN-DM (Summ) |
|---|---|---|
| AR Transformer (Vaswani et al., 2017) | 27.3 | 36.8 |
| SUNDAE (Savinov et al., 2022) | 26.3 | 37.0 |
| CMLM (Ghazvininejad et al., 2019) | 24.6 | — |
| Levenshtein Transformer[2] (Gu et al., 2019) | 23.7 | — |
| DisCo (?) | 24.7 | — |
| Imputer | 25.2 | — |
| DIFFUSER | 27.2 | 37.8 |
| DIFFUSER + AR bootstrap | **28.8** | 38.4 |
| DIFFUSER + source bootstrap | 24.5 | **38.9** |

Table 1: Machine Translation (MT) and Summarization (Summ) results on WMT'14 En-De (gold) and CNN-DailyMail. Experiments on MT use BLEU while summarization uses ROUGE. DIFFUSER is compatible with a standard autoregressive model, while outperforming previous methods.

| Model | Accuracy | BLEU |
|---|---|---|
| Masker (Malmi et al., 2020) | 40.9 | 14.5 |
| Tag and Generate (Madaan et al., 2020) | 86.2 | 19.8 |
| LEWIS (Reid & Zhong, 2021) | **93.1** | 24.0 |
| DIFFUSER | 87.6 | **25.2** |

Table 2: Results on Yelp dataset for text style transfer. Without task-specific training techniques, DIFFUSER performs comparably to previous task-specific methods.

source-target pairs) leading to the prominence of unsupervised text style transfer methods. We propose a method of performing unsupervised text style transfer using DIFFUSER, following the synthetic generation method in Reid & Zhong (2021). We train two separate, style-specific (e.g. positive and negative) DIFFUSER models on the style-specific data. We then perform transfer at test time, feeding text from each style into the model trained to edit in the opposite style (e.g. positive text → negative DIFFUSER model; negative text → positive DIFFUSER model). Following standard practice, we measure performance with BLEU, Self-BLEU and Accuracy (based on a classifier trained to disambiguate between different styles of text; we use the classifier from Reid & Zhong (2021)).

## 4.3 RESULTS

**Main Results** We summarize our main results on both machine translation and summarization in Table 1. As can be seen, for both machine translation and summarization tasks, DIFFUSER, using 12 diffusion steps, outperforms all non-autoregressive baselines[1] and rivals or outperforms the fully autoregressive model. Particularly interesting is how the various methods of initializing our model (i.e. AR Bootstrap and Source Bootstrap) can further improve performance well beyond the autoregressive baseline, depending on the task. We can see that for summarization, bootstrapping from the source input is more effective than bootstrapping from an abstractive autoregressive model. However, for both tasks, unlike many non-autoregressive methods, we show that DIFFUSER is complementary with token-level autoregressive methods and can be used naturally in conjunction with them.

**Style Transfer Results** We also perform unsupervised text style transfer using our DIFFUSER models using the Yelp (Shen et al., 2017) dataset. The results can be seen in Table 2. We show that even without task-specific techniques (such as synthetic data generation and classifier based style-specific token identification), we still have competitive performance with state of the art methods.

## 4.4 ANALYSIS

---

[1]We were not able to reproduce the published results of the Levenshtein Transformer using their code, hence our reported BLEU score of 23.7 is slightly lower than that of 25.2 reported in Gu et al. (2019)

| | |
|---|---|
| Source Document | (CNN)They're not gonna take it anymore. Really. Twisted Sister says that its 2016 tour will be its last, according to a press release. Next year marks the band's 40th anniversary, and to celebrate, the tour is being titled "Forty and F*ck It." "It's official: Farewell," Twisted Sister singer Dee Snider posted on Facebook. Snider also noted that the band will play with a new drummer, Mike Portnoy of Adrenaline Mob. Portnoy replaces A.J. Pero, who died March 20. The band will also perform two shows in Pero's honor: one at Las Vegas' Hard Rock Hotel and Casino, the other at the Starland Ballroom in Sayreville, New Jersey. The latter is in support of Pero's family. Twisted Sister's biggest hit, "We're Not Gonna Take It," hit the Top Forty in 1984 and was featured in a popular video. |
| Step 1 | ~~(CNN)They're not gonna take it anymore. Really.~~ Twisted Sister says that its 2016 tour will be its last, according to a press release. Next year marks the band's 40th anniversary, and to celebrate, the tour is being titled "Forty and F*ck It." "It's official: Farewell," Twisted Sister singer Dee Snider posted on Facebook. Snider also noted that the band will play with a new drummer, Mike Portnoy of Adrenaline Mob. Portnoy replaces A.J. Pero, who died March 20. The band will also perform two shows in Pero's honor: one at Las Vegas' Hard Rock Hotel and Casino, the other at the Starland Ballroom in Sayreville, New Jersey. The latter is in support of Pero's family. Twisted Sister's biggest hit, "We're Not Gonna Take It," hit the Top Forty in 1984 and was featured in a popular video. |
| Step 2 | Twisted Sister says that its 2016 tour will be its last, according to a press release. Next year marks the band's 40th anniversary, and to celebrate, the tour is being titled "Forty and F*ck It." ~~"It's official: Farewell," Twisted Sister singer Dee Snider posted on Facebook. Snider also noted that the band will play with a new drummer, Mike Portnoy of Adrenaline Mob.~~ Portnoy replaces A.J. Pero, who died March 20. The band will also perform two shows in Pero's honor: one at Las Vegas' Hard Rock Hotel and Casino, the other at the Starland Ballroom in Sayreville, New Jersey. The latter is in support of Pero's family. Twisted Sister's biggest hit, "We're Not Gonna Take It," hit the Top Forty in 1984 and was featured in a popular video. |
| Step 3 | Twisted Sister says that its 2016 tour will be its last, according to a press release. Next year marks the band's 40th anniversary, and to celebrate, the tour is being titled "Forty and F*ck It." Portnoy replaces A.J. Pero, who died March 20. The band will ~~also~~ perform two shows in Pero's honor ~~: one at Las Vegas ' Hard Rock Hotel and Casino, the other at the Starland Ballroom in Sayreville, New Jersey. The latter is in support of Pero's family. Twisted Sister's biggest hit, "We're Not Gonna Take It," hit the Top Forty in 1984 and was featured in a popular video~~ in Las Vegas and New Jersey. |
| Step 4 | Twisted Sister says that its 2016 tour will be its last, according to a press release. Next year marks the band's 40th anniversary, and to celebrate, the tour is being titled "Forty and F*ck It." ~~Portnoy replaces~~ A.J. Pero, ~~who~~ died March 20. The band will perform two shows in Pero's honor in Las Vegas and New Jersey. |
| Generated Summary | Twisted Sister says that its 2016 tour will be its last. Next year marks the band's 40th anniversary, and to celebrate, the tour is being titled "Forty and F*ck It." A.J. Pero, died March 20. The band will perform two shows in Pero's honor in Las Vegas and New Jersey. |

Table 4: Example of our summarization DIFFUSER process on a test set example. Here we show that the majority of the summarization process is deletion coupled with minor edits. Despite this simplicity, we are able to improve over existing purely abstractive models.

We perform additional analyses on DIF-FUSER, specifically focusing on the decoding method, the number of iterations versus the final BLEU score, and also a qualitative analysis of how text changes at every step.

| Initialization | Decoding Method | BLEU |
|---|---|---|
| Random Tokens | Greedy | 26.3 |
| Random Tokens | Beam $b = 5$ | 26.7 |
| Random Tokens | Beam $b = 15$ | 26.9 |
| Random Tokens | Nucleus | 26.8 |
| Random Tokens | 2D-Beam | 27.2 |

Table 3: Decoding method ablation on the MT test set.

**Decoding Method Ablation** We perform an ablation of the decoding method, using DIFFUSER for 12 steps (as used in our main results) and showing results when comparing greedy decoding, (1D) beam search, nucleus decoding, and 2D beam search. We show that 2D-beam search tends to perform the best, likely because it searches over multiple diffusion steps, while other methods (greedy, beam, nucleus) are still competitive.

**Number of Edit Steps versus Performance** We perform an analysis where we compare the number of timesteps in our denoising diffusion process and the final BLEU score on WMT'14 En-De when using 2D-Beam Search and random token initialization in Figure 4. Here it can be seen that most performance gains are in the initial diffusion timesteps (0-10), with diminishing gains (for machine translation) or gradual losses (for summarization) between 10 and 30, after which performance marginally decreases towards 60 steps.

**How does text change every step?** We include a qualitative sample from our DIFFUSER summarization model (Table 4). We find that DIFFUSER learns edit processes intuitive to the task at hand: namely largely deleting portions and making minor edits to the remaining text (similar to how a human may perform summarization given a news article).

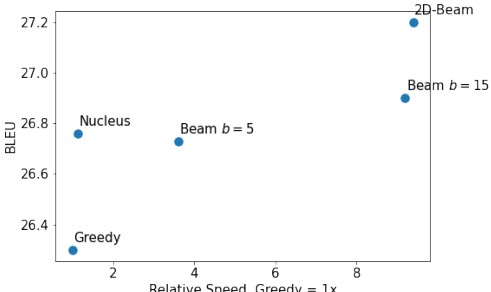 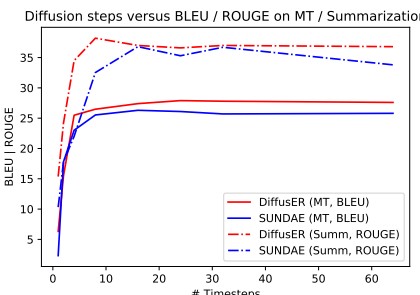

Figure 3: Relative time (seconds) comparison between decoding methods, measured on a single V100 GPU. There is a trade-off between inference cost and performance. Faster well-performing decoding algorithms for diffusion models are an area for further work.

Figure 4: Number of steps versus BLEU/ROUGE on WMT'14 En-De and Summarization for both SUNDAE and DIFFUSER. We observe fast initial progression with performance, leveling off as steps increase.

**Time comparsion between decoding methods** We also measure the impact of the various decoding algorithms we used with results shown in Figure 3. Beam search and 2D-Beam Search performs significantly slower than greedy and nucleus sampling, demonstrating the potential for improved decoding algorithms tailored for improving the trade-off between efficiency and accuracy in diffusion models.

## 5 RELATED WORK

**Non-Autoregressive Generation** Work in machine translation has explored non/semi-autoregressive generation (Gu et al., 2017; Lee et al., 2018), which often includes an iterative refinement step (Lee et al., 2018; Ghazvininejad et al., 2019; Kasai et al., 2020a; Gu et al., 2019). Previous methods in this space are often highly specialized underperform non-autoregressive methods due to the constraints imposed on generation for efficiency. This being said, Kasai et al. (2020b) demonstrated that non-autoregressive models are actually comparable in speed when using a larger batch size instead of 1. Our method allows us to hone in on the notion of iterative refinement by way of editing processes, and is also relatively general, allowing us to combine DIFFUSER with standard autoregressive models.

**Learning Properties of Edits** Previous work has also looked at studying or exploiting the properties of edits. This was initially worked on in the context of vector representation learning of edits (Yin et al., 2019; Marrese-Taylor et al., 2021). Concurrently, a line of work has used edits for specific tasks such as sentence fusion, style transfer and grammatical error correction (Malmi et al., 2019; 2020; Reid & Zhong, 2021; Omelianchuk et al., 2020). Recent work has proposed *editing processes* (Reid & Neubig, 2022), in which document generation is looked at through the lens of its revision history, rather than just at a token level. We take inspiration from this work and devise a process by which arbitrary text generation tasks can be fitted into this framework.

## 6 CONCLUSIONS

We proposed DIFFUSER, an diffusion-based generative model for text using edits. DIFFUSER shows improvements across the tasks considered (machine translation, summarization, style transfer), with improved generative flexibility via incremental text improvement, and compatibility with standard autoregressive models. We hope that DIFFUSER with spur research on edit-based generative models, with further potentials including how we can leverage edits to ensemble models (regardless of parameter count) in the discrete space.

## ACKNOWLEDGEMENTS

We thank Armen Aghajanyan, Daniel Fried, Edison Marrese-Taylor, Eric Wallace, and Luke Zettlemoyer for their helpful comments in early discussions. We thank Ari Holtzman, Jungo Kasai, Aman Madaan, and Eric Wallace for feedback and proofreading the draft of this paper.

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
