# OpenReview forum: "DiffusER: Diffusion via Edit-based Reconstruction"
_ICLR.cc/2023/Conference — ICLR 2023 poster_

### Official Review · Reviewer_E15v · 2022-10-25

**Confidence:** 3
**Correctness:** 3
**Technical Novelty And Significance:** 2
**Empirical Novelty And Significance:** 2
**Recommendation:** 6

**Clarity, Quality, Novelty And Reproducibility:**

The main approaches and contributions of the paper is clearly presented and well organized. However, more clarity is needed on some details in the descriptions of models and experiments. It did not affect the claim of the paper, though.

The approaches and claims are solid and sound, backed up by experiment results.

There is novelty in the approach, but it is not considered significant, since there are already existing work on NLP diffusion models as described in Sec 2.3.

The authors stated that "Code and data to reproduce experiments will be released".


**Strength And Weaknesses:**

Pros:
- Well motivated, provided valuable exploration in the recent popular direction of diffusion models in NLP area.
- Very effective diffusion approach in NLP by adopting flexible editing operations.
- Strong results in experiments on multiple tasks and datasets, compared against strong baselines including recent literature on non-auto-regressive generation models.
- Effective techniques improved on top of the diffusion approach, including bootstrapping and 2D-beam. Ablation and analysis are also provided.

Cons:
- There is novelty in the approach, but it is not considered significant, since there is already existing work on NLP diffusion models as described in Sec 2.3.
- The description of models and experiment settings is not clear enough. Table 4 could be moved to Appendix to make room for clearer and more detailed descriptions and explanations.
  1. Since the tasks are sequence to sequence generation, the Diffuser model can be looked on as the decoder, should there also be an encoder to take in the input text, so that the decoder can condition the output on the input? This was not described anywhere in the paper.
  1. How many diffusion steps, and what b and r in Summarization and Text Style Transfer tasks?
  1. Equation (3) is not explained clearly, including some symbols not annotated.
  1. There is no description of the Accuracy metric in Table 2.
- It would be helpful if some experiments and analysis are included for the number of diffusion steps in *training*.
- Numbers of training time and decoding time compared to baselines would be helpful, together with some discussion on the trade-off of efficiency and efficacy.

Minor edit:
Page 8: "comparsion" -> "comparison"


**Summary Of The Paper:**

The paper proposes DIFFUSER, a denoising diffusion model for text generative tasks. It treats text generation as a Markov chain of Levenshtein edit steps to denoise from the initial text. An editing step is modeled as an editing process in existing work (Reid & Neubig).

The contribution of the paper is mainly 1) using the more flexible edit operations in diffusion models; 2) more thoughtful bootstrapping from autoregressive output or source text as initial text sequence; 3) employing 2D-beam search during decoding.


**Summary Of The Review:**

The paper is well-motivated and provided valuable exploration in the recent popular direction of diffusion models in NLP area. The approach, although not strong in novelty, effectively employed various techniques to obtain strong experiment results on multiple tasks. The presentation is mostly clear and well organized, although there are multiple places that need to be clarified or further explained and discussed.

---

> ### Author Response · Authors · 2022-11-17
> **Response**
>
> We thank the reviewer for their comments and feedback. We’re glad that you found our work to be well-motivated + presented, our method to be effective, and experiments solid. We have addressed the mentioned concerns below and hope our response assuages your concerns.
>
> > Since the tasks are sequence to sequence generation, the Diffuser model can be looked on as the decoder, should there also be an encoder to take in the input text, so that the decoder can condition the output on the input? This was not described anywhere in the paper.
>
> We agree that we could state this more explicitly, and we have made the corresponding changes in revision.  We mention this briefly in Section 4.1 as “Transformer-base”, an encoder-decoder model.
>
> > How many diffusion steps, and what b and r in Summarization and Text Style Transfer tasks?
>
> As stated in Section 4.2, we set b = 5 and r = 3.
>
> > There is no description of the Accuracy metric in Table 2.
>
> We use the classifier used by Reid and Zhong, 2021. We have revised the text in Section 4.2 to reflect this.
>
> > Equation (3) is not explained clearly, including some symbols not annotated.
>
> Thank you for pointing this out! We have added extra context in Section 2.2, given background in Section 2 of Reid and Neubig, 2022
>
> > It would be helpful if some experiments and analysis are included for the number of diffusion steps in training.
>
> We use a zero-th order Markov chain, meaning that we model diffusion steps individually, regardless of their step in the synthetic corruption process. However, if you are referring to the number of corruption steps that we use to generate training data, we use four steps (given that the process of keeping 60% of tokens four times results in the probability of any original token remaining to be ≈ 0.025)using the sampling probabilities outlined in the paper.
>
> > Discussion on efficiency and efficacy
>
> We note that the main drawback of this approach right now is speed (especially with 2D beam search), but work happening in diffusion models on the vision side have shown that significant speedups are often possible with further research (such as the work in Kong and Ping, 2021; https://arxiv.org/abs/2106.00132). Due to space constraints, we are not able to include this in the main text, but will include this in revision.

---

### Official Review · Reviewer_Gnic · 2022-10-26

**Confidence:** 4
**Clarity, Quality, Novelty And Reproducibility:** See S&W section.
**Correctness:** 1
**Technical Novelty And Significance:** 3
**Empirical Novelty And Significance:** Not applicable
**Recommendation:** 8

**Strength And Weaknesses:**

Strengths

- The model is intuitively appealing, and the paper in general is well written.
- The experiments demonstrate competitive performance, and the advantages of being able to bootstrap and refine results.

Limitations

- While the paper well written, and establishes the approach well, it could benefit tremendously from some additional supplementary material, to give the paper more depth, as discussed in the following points.
- The work is well executed and the model intuitive, but in context, an obvious next step given current work on iterative text generation models, and the recent success of diffusion models. More credit should be given to previous related work that establishes similar reconstruction and corruption processes, similar two-stage editing, and similar decoding processes. These are not novel components of this paper as suggested by the current manuscript. Related, I'd like the authors, to the extent possible, to discuss the similarities/differences advantages/disadvantages over similar text editing models, such as the Levenshtien transformer. Why would it not perform as well as Diffuser?
- More flexible control over generation is claimed as an advantage of the model, but this is really never demonstrated or exercised. While I can imagine variations of the model that could deliver this, the models investigated are trained and utilized end-to-end, with intermediate generations that have no notion of validity associated with them, and internal decisions that, if manipulated, would likely degrade performance.
- Related, investigating the generalization of and effects of manipulating the editing priors at test time to exercise some control and/or diversify outputs would be an interesting ablation/demonstration.
- Statements like "editing processes can also be used to calculate the probability of only the final document while taking into account previous revisions, which is not possible in the traditional text generation setup", are misleading. Under the model intermediate revisions are meaningless, and conventional LMs can evaluate likelihood in a single pass (i.e., having to go through a revision process for likelihood evaluation is a disadvantage of the model). Generally speaking, a more balanced and frank presentation of the strengths and limitations of the approach, properly situated within context with previous work, would improve the paper signficantly.
- Equation 5 has errors, please correct.

**Summary Of The Paper:**

Motivated by how humans revise content and the success of diffusion models with continuous inputs, the authors propose a generative model of text based on multiple editing steps, with each step based on one or more text span editing operations (insert, delete, replace, keep). The generative model is trained to invert, step by step, the random edits of a "text diffusion process", which is specified by a prior over edits and edit length. Generation at each step is further decomposed into a model for generating the edit operations to apply at each position of the current input text, and a model for generating text given the input and the selected edit operations, similarly to many existing edit-based generation approaches.

Results on machine translation (WMT 14') and summarization (CNN Daily Mail) show competitive performance (Table 1). An advantage of the proposed Diffuser model can refine text and so can be boostrapped by an autoregressive translation or the source to summarize, which boosts performance (Table 1). The authors also show that by simply training 2 models for positive and negative sentiment Yelp reviews and then using them to transform the input text to the target sentiment, performance is competitive with SOTA task specific models, which is impressive. Some ablations around decoding method vs. speed and performance and seed text type are also included.

**Summary Of The Review:**

Overall, the presented text-edit based diffusion model is much anticipated, well motivated, and has been well executed. However, as detailed in the limitations section, more credit and context as it relates to previous work should be included when presenting the elements of the model, and both the strengths and limitations of the approach should be frankly discussed, in supplementary material if necessary, to give the paper more context and depth. If feasible, additional experiments that investigate the extent that generations can be flexibly controlled would further strengthen the paper.

---

> ### Author Response · Authors · 2022-11-17
> **Response**
>
> We thank the reviewer for their comments! We appreciate that the reviewer finds our work to be significant and original, and experiments to be comprehensive.
>
> > It would be nice to consider more post-editing-like baselines, e.g., just adding some off-the-shelf post editors or grammatical-error correction models to the MT output.
>
> We compared with LEWIS, which we believe reflects this notion of a post-editing style model to some extent for style transfer. If you have suggestions for other, general purpose, off-the-shelf editors (with usable code) we would be happy to consider comparing with them!
>
> > While sensible, the training objective is a bit limited in that it denoises pure random tokens. Concretely, to get good performance at test time, the model needs to denoise generations that have various errors spanning semantic errors, syntactic inconsistencies, etc. These types of errors are quite far from swapping in random tokens in the input.
>
> We think that it is promising that DiffusER already gets performance with random noise, and believe that future methods that introduce errors in more sophisticated ways may further improve performance. It is with noting, however, that DiffusER can successfully improve outputs generated by autoregressive models/other outputs by simply using this naive noise-based signal.

---

### Official Review · Reviewer_MpCU · 2022-10-31

**Confidence:** 4
**Correctness:** 3
**Technical Novelty And Significance:** 3
**Empirical Novelty And Significance:** 4
**Recommendation:** 8

**Clarity, Quality, Novelty And Reproducibility:**

The paper is very clearly written. The work is quite original. The overall experiment evaluations are quite high quality and comprehensive.

**Strength And Weaknesses:**


Strengths:
* The objective and model design is quite novel and refreshing. I appreciate the careful design of the architecture---rather than focusing on a pure end-to-end system is decomposes the task into edit tagging and generation.
* The training objective seems sensible, and the overall evaluation is pretty solid. I specifically appreciate using the model as a sort of "general-purpose post-editing" model (as shown in Diffuser + AR bootstrap). It would be nice to consider more post-editing-like baselines, e.g., just adding some off-the-shelf post editors or grammatical-error correction models to the MT output.

Weaknesses:
* While sensible, the training objective is a bit limited in that it denoises pure random tokens. Concretely, to get good performance at test time, the model needs to denoise generations that have various errors spanning semantic errors, syntactic inconsistencies, etc. These types of errors are quite far from swapping in random tokens in the input.
* The editing-based evaluation is a bit limited. It would have been great to explore more some of the capabilities/failures of the model. For example, conditioning on various keywords on the target side, conditioning on a target syntactic style, or similar evaluations.

**Summary Of The Paper:**

This paper studies an edit-based generative text model that starts with a complete noise distribution (random gibberish) as input and then produces a series of edits to reach high-quality output. Inspired by diffusion models in CV, this Diffuser model rivals or outperforms standard autoregressive models on various generation tasks (MT, summarization), while also providing additional editing-based functionality.

**Summary Of The Review:**

I think methods for non-traditional forms of generating text are widely underexplored, and this paper proposes a method that rivals typical autoregressive generation methods that are commonplace today. The overall writing and paper quality is solid.

---

> ### Author Response · Authors · 2022-11-17
> **Response to RMpCU**
>
> We thank the reviewer for their feedback and glad that they find our approach to be novel, and carefully thought through. We look to address any concerns in the text below:
> > Why would the Levenshtein Transformer not perform as well as Diffuser?
>
> The Levenshtein transformer is trained with on the fly dynamic programming (which doesn’t allow for deviation from the final goal/the inductive bias of having multi-step edits) and performs different types of edits (replace/delete) in multiple steps. Whereas we use this multi-order corruption diffusion process which allows for a unified process wherein all types of edits can be performed simultaneously while allowing for the model to learn the inductive bias to generate incorrect text (e.g. with the relationship between our delete/insert operations) -> then self-correct.
>
> > Under the model intermediate revisions are meaningless, and conventional LMs can evaluate likelihood in a single pass (i.e., having to go through a revision process for likelihood evaluation is a disadvantage of the model).
>
> Although this is the case from an initial perspective, this notion of iterative refinement can help the model directly model the notion of improvement, which could help for further improvements over conventional LMs. The main drawback of this approach right now is speed, but work happening in diffusion models on the vision side have shown that this can be mitigated to some extent with certain techniques and tricks.
>
> > Equation 5 has errors, please correct.
>
> We have corrected the errors, thank you for pointing this out.

---

### Author Response · Authors · 2022-11-17
**Response to all reviewers**

We thank the reviewers for their feedback. We are glad that they find our paper to be significant and original (RE15v, RGnic) and method to be novel and carefully thought through (RMpCU), while being well motivated and presented (RE15v). We have addressed individual concerns in the individual replies below, and updated the paper (highlighting added text in blue). Thank you for helping us strengthen our paper!

---

### Decision · Program_Chairs · 2023-01-20

**Decision:**

Accept: poster

**Justification For Why Not Higher Score:**

N/A

**Justification For Why Not Lower Score:**

It's a good paper

**Metareview: Summary, Strengths And Weaknesses:**

The idea and the method proposed in this paper is novel and all reviewers appreciated the model design. The method can be a general purpose edit method. The evaluation seems solid although some reviewers found it limited and could be compared with other edit-based approaches like GEC (MpCU). Reviewer Gnic also recommended to describe its similarity/dissimilarity and advantages/disadvantages with other similar method. Overall, considering its strength, I am in favour of accepting the paper.

**Note From Pc:**

if the above contains the word "oral" or "spotlight" please see: "oral" presentation means -> notable-top-5% and "spotlight" means -> notable-top-25%. As stated in our emails, we are disassociating presentation type from AC recommendations

**Summary Of Ac-Reviewer Meeting:**

N/A